# Evaluation of a ‘Preventative’ Strategy to Manage Spider Mites on Almond

**DOI:** 10.3390/insects11110772

**Published:** 2020-11-09

**Authors:** Kristen Tollerup, Bradley Higbee

**Affiliations:** 1Kearney Agricultural Research and Extension Station, Agriculture and Natural Resources, University of California, Parlier, CA 93648, USA; 2Trécé Inc., Adair, OK 74330, USA; bhigbee@trece.com

**Keywords:** pacific mite, acaricides, abamectin, resistance, sixspotted thrips

## Abstract

**Simple Summary:**

The almond industry in California produces approximately 80% of the world’s supply and currently consists of roughly 0.45 million bearing and non-bearing hectares. Spider mite management is a component of annual arthropod management strategies on almond. Regularly, acaricides containing the active ingredient, abamectin, are applied in spring when mites are not yet present or well below a level which justifies treatment, in other words applied to prevent rather than treat existing populations. Although other acaricides are used, those containing the active ingredient, abamectin, accounted for 68% to 95% of spring (preventative) applications between 2005 and 2017. Although the strategy is widely employed in the almond industry, there are no published studies supporting the practice. Moreover, the strategy runs contrary to sustainable, integrated pest management practices by forgoing monitoring, reliance on spider mite natural enemies, and economic threshold levels. In this study, we conducted large field trials to evaluate the effectiveness of preventative acaricide applications. Results showed that farmers can forego a preventative strategy since spider mite densities typically remain well below damaging levels through the period critical for tree growth and nut yield. Also, results showed that such a heavy reliance on abamectin has led to a low to moderate level of resistance developing in some spider mite populations in the southern and mid San Joaquin Valley.

**Abstract:**

Field experiments were conducted in two commercial almond orchards located in the southern San Joaquin Valley during 2016 and 2017 to evaluate a “preventative” strategy to manage spider mites. Pacific mite, *Tetranychus pacificus* McGregor, was identified as the only mite species infesting the experimental sites in both years. We monitored mites weekly in 3.6-hectare plots over approximately 21 weeks in 2016 and in 2017 using guidelines developed by the University of California. In late May, prior to the detection of mites, preventative acaricide treatments, abamectin, cyflumetofen, or etoxazole were applied to the experimental plots at a field rate. In 2016 and 2017, mite densities in all the treatments increased at early-July, peaked at mid-August, and were undetectable by late August. Preventative acaricide-treated plots in 2016 tended to have significantly lower mite densities than in the untreated control plots. Although in 2017, densities in the acaricide-treated plots tended to not significantly differ from control plots. Mite feeding injury, measured as mean cumulative mite-days, did not exceed the economic threshold during the experiment. The biological control agent, sixspotted thrips, *Scolothrips sexmaculatus* (Pergande) likely played a role in controlling mite populations at mid and late August. Our results indicate that a preventative strategy does not play a definitive role in *T. pacificus* management on almond. Additionally, acaricides with the active ingredient, abamectin, are heavily relied on as preventative treatments. We assessed populations of *T. pacificus* from the mid and southern San Joaquin Valley and found increased tolerance to a medium level of resistance to the acaricide.

## 1. Introduction

Almond production occurs throughout California’s Central Valley and consists of approximately 0.45 million bearing and non-bearing hectares. The industry plays an important economic role generating a crop value of ~5.60 billion dollars (US) in 2016, ranking third behind only dairy and grape [1]. Two spider mite species attack almond in California, the twospotted spider mite, *Tetranychus urticae* Koch, occurring predominately in the northern San Joaquin Valley and Sacramento Valley region, and the Pacific mite, *Tetranychus pacificus* McGregor, occurring predominately in the mid and southern valley [2,3]. 

Spider mites feed by piercing cells at the epidermal layer of leaves using a stylet mouthpart then sucking out the cell contents. Feeding injury reduces mesophyll and stomatal conductance and decreases photosynthesis [4]. Barnes and Andrews [5] determined that on almond a significant reduction in vegetative and nut formation occurred after approximately 424 mite-days, defined as one mite feeding for one day. Additionally, heavy infestations on almond late in the season, exacerbated from activities associated with harvest, have a high potential for causing defoliation resulting in limb and trunk sun burn and inducing trees to re-flush.

The University of California Statewide Integrated Pest Management (IPM) Program *Almond Pest Management Guidelines* recommends monitoring for mite motiles and eggs, and spider mite natural enemies such as phytoseiid mites and sixspotted thrips from May through August. The guidelines specify that a miticide application is not justified until approximately 32% and 53% of leaves are infested in the absence or presence of predators, respectively. The practice, however, of applying a preventative acaricide treatment in spring when spider mites are absent or well below the treatment threshold has become common in the almond industry.

From 2005 to 2017, the California Department of Pesticide Regulation (CDPR) reported a 3-fold increase in the number of hectares treated in spring [6]. Acaricides containing the active ingredient, abamectin, are preferred for spring applications. The compound has translaminar activity, providing protection against rapid photodegradation and extending its residual activity [7]. Moreover, CDPR indicated that the acaricide accounted for between 68% to 95% of spring applications. The heavy reliance on abamectin and the ability of spider mites for developing resistance [8], likely has led to some spider mite populations becoming resistant. In 2013 farmers and pest control advisers in the almond industry reported lower efficacy of acaricides, with the active ingredient, abamectin [9].

In this study, we determined if a preventative acaricide application of abamectin, cyflumetofen, or etoxazole provided a benefit for managing spider mites on almond by suppressing economically damaging populations up to the hull-split phase of growth. Additionally, we conducted bioassays to evaluate *T. pacificus* for resistance to abamectin using mites collected from our experimental sites and orchards located in the mid and southern San Joaquin Valley. 

## 2. Materials and Methods 

### 2.1. Effectiveness of Preventative Acaricide Applications 

We established plots approximately 3.6-hectare in size consisting of 17 tree-rows × 36 trees at a spacing of 6.7 × 5.5 m. Plots were established at each of two, 61-hectare commercial almond orchards located near Shafter, Kern County and Wasco, Kern County. Both orchards contained the cultivars, Nonpareil and Sonora [*Prunus dulcis* (Mill.) D.A. Webb]. Treatments consisted of an untreated control, Abamectin at 26.3 g AI/ha (ABBA 0.15 EC, Makhteshim Agan of North America Inc.) plus alcohol ethoxylate at 3.3 L/ha (Vintre, Oro Agri, Fresno, CA, USA), cyflumetofen at 200 g AI/ha (Nealta, BASF, Triangle Park, NC, USA), and etoxazole at 151 g AI/ha (Zeal, Valent, Walnut Creek, CA, USA).

In 2016 we used an unbalanced design consisting of eight untreated control plots and six plots for each acaricide treatment. In 2017, we used a balanced design consisting of six plots for the untreated control and each of the acaricide treatments. The commercial grower applied treatments using a commercial engine-driven air-blast sprayer traveling at 3.2 km/h and 1874 L of water per hectare. The grower applied treatments at late May during 2016 and 2017. In 2017 experimental plots received the same treatment as in 2016.

We assessed spider mite densities weekly or bi-weekly from May to early October (2016) and mid-May to mid-September (2017). Within plots, we established nine (2016) or seven (2017) sampling areas, each consisting of 25 trees (5 rows × 5 trees). We established sample areas located at the plot center and equally spaced along the plot’s edges. To reduce treatment cross contamination, a buffer of at least 28 m existed between sampling areas of adjacent plots. On sampling dates, we randomly-selected a single tree per sampling area and collected 15 leaves, approximately 1.8 m above the soil surface from around the tree on the inner and outer portion of the canopy. Leaf samples from each tree were placed into a labeled 0.94 L, Sunny Select, re-closeable plastic bag (Super Store Industries, Lathrop, CA, USA). Plastic bags were immediately placed into an ice chest to chill the sampled leaves and minimize the movement of motile mite stages among the leaves. Until processed, leaf samples were stored at approximately 7 °C.

In the laboratory, we examined the abaxial and adaxial side of each leaf utilizing a binocular dissecting microscope. For each 15-leaf sample, the number of eggs as well as adults and immature stages of spider and phytoseiid predator mites were counted. At the initiation of the study, 10 adult males were collected, slide mounted, and identified using the morphological characteristic of the aedeagus [10]. Also, we examined leaves for the presence of sixspotted thrips, *Scolothrips sexmaculatus* (Pergande) and other generalist insect predators. Sixspotted thrips were identified according to Mound [11].

### 2.2. Natural Enemy Sampling

During 2017, we monitored for *S. sexmaculatus* using 8.9 × 20.3 cm, yellow/blue sticky cards (Alpha Scents, West Linn, OR, USA). A single card was placed at approximately 1.8 m above the soil surface in the canopy of a tree. We monitored two plots of each treatment at each of the experiment orchard sites and changed cards weekly or biweekly.

### 2.3. Spider Mite Colony to Determine Base-Line Susceptibility to Abamectin 

A colony of *T. pacificus* was established, using individuals obtained from a susceptible population maintained in the Frank Zalom laboratory at the University of California, Davis. The susceptible *T. pacificus* was maintained at the University of California, Kearney Agricultural Experiment and Extension Center, Parlier, on bean, *Phaseolus vulgarus* (L.) under a 12:12 L, D photo period, and 30 °C.

### 2.4. Collection of Field Populations for Bioassays

Between mid-May and early June of 2017, we monitored 15 almond orchards bi-weekly located in Tulare County, southern and mid-Fresno county, and northern Kern County for spider mite populations. Sampling was conducted by selecting five trees along the orchard’s edge, selecting 15 leaves per tree and examining abaxial leaf surface with a 10× hand loupe. Spider mite infestations began occurring at early June and samples were collected from seven of the orchards and the two experimental sites between 4 June to September 2017. Mite-infested leaves were collected by cutting branches approximately 10 mm in diameter from the interior area of five to 10 trees. Excised branches were placed in 3.79-L, Ziploc plastic bag (SC Johnson and Son, Inc, Racine, WI, USA) and kept in an ice chest until arriving at the laboratory. Sampled mites were maintained in the laboratory on the sampled branches and kept for no longer than 72 h before conducting the bioassays.

### 2.5. Laboratory Bioassays

For each bioassay experiment, we evaluated four concentrations of abamectin (Agri-Mek SC, 8% ai [ai wt/v] Syngenta Crop Protection, LLC, Greensboro, NC, USA) in water plus an untreated control. A stock solution (65.5 ppm) was prepared, and test concentrations were prepared using serial dilutions. 

A replicate consisted of 10 mature females placed on an 18-mm leaf disc cut from a mature almond leaf. On the same date as a bioassay experiment, almond leaves were collected from an orchard not treated with acaricides located at the Kearney Agricultural Center, Parlier. Leaves were collected, washed with distilled water, and discs cut. Leaf discs were placed on paper towels and approximately 12 µL of abamectin dilutions applied using a PREVAL^®^ sprayer (Chicago Aerosol, Coal City, IL, USA). Six to 10 replicates were used for each bioassay trial.

The treated leaf discs were allowed to dry for approximately one hour then placed on moist cotton contained in a 150 × 15 mm (diameter × height) plastic Petri dish arena. Ten adult females were placed on each leaf disc using a small camel-hair paint brush. Arenas were kept in a designated area of a laboratory workspace and kept at 25 °C and 30% RH. Mite mortality was scored after 24 h of exposure and mites were considered dead if paralyzed and unable to ambulate at least one body length after prodding. 

### 2.6. Statistical Analyses

Analysis for the main effect, treatment, was conducted using a repeated measures mixed model (PROC MIXED) based on SAS statistical software [12]. The mean number of all stages and motile stage only per leaf by sample date were analyzed with plots designated as the subjects. Least squares-means were computed using the lsmeans option for the treatment effect and compared using *t*-test pair-wise comparisons among the treatments using the pdiff option with Tukey’s adjustment. The pdiff option requests *t*-test *p*-values for the differences of the lease square-means while the Tukey option adjusts *p* values based on the number of multiple comparisons calculated. 

Additionally, the mean cumulative number of mite-days were calculated for each treatment by multiplying the mean number of mites per leaf, at successive sample days, by the sample interval (days) and adding the number of mite days accumulated throughout the experiment period. To determine statistical differences among the cumulative mean mite-days, we utilized the non-parametric (PROC NPAR1WAY) procedure base on the SAS statistical software [12]. Multiple non-parametric pair-wise comparisons were conducted using the Dwass, Steel, Critchlow-Fligner [13] option.

The (PROC PROBIT) procedure, based on SAS statistical software [12], was used to analyze response i.e., mortality to abamectin. The log base 10 option was utilized for the dose response and confidence limits were calculated using the inversecl option. Resistance ratios (RR) were calculated as (LC_50_ field population)/(LC_50_ susceptible population) along with 95% confidence intervals calculated according to Robertson et al. [14].

## 3. Results

### 3.1. Effectiveness of Preventative Acaricide Applications 

*Tetranychus pacificus* accounted for the only spider mite species identified at each of the experimental sites during both 2016 and 2017. Motiles plus eggs consistently remained below 0.1 per leaf for much of the experiment, 8 May to 15 July, and 29 August to 8 October of 2016 (Figure 1); and 15 May to 10 July, and 30 August to 11 September of 2017 (Figure 2). Mite density in all treatments did not reach the *Almond Pest Management Guidelines* threshold of two (all stages) per leaf until approximately seven to 10 days prior to harvest (Figure 1 and Figure 2). 

In 2016, the preventative acaricide treatment, cyflumetofen had significantly lower densities of all mite stages than the control on 25 July (*df* = 22, *t* = 2.85, *p* = 0.0439). The density of all mite stages was significantly lower than the untreated control on 1 August than in the abamectin (*df* = 22, *t* = 4.44, *p* = 0.001), cyflumetofen (*df* = 22, *t* = 4.39, *p* = 0.001), and the etoxazole (*df* = 22, *t* = 4.67, *p* = 0.001) treatments. Similarly, on 8 August the abamectin (*df* = 22, *t* = 4.07, *p* = 0.0027), cyflumetofen (*df* = 22, *t* = 4.42, *p* = 0.001), and etoxazole (*df* = 22, *t* = 4.95, *p* < 0.001) treatments each had significantly lower densities than the untreated control (Figure 1). During 2017, on 21 August only abamectin (*df* = 20, *t* = 3.45, *p* = 0.0124) and etoxazole (*df* = 20, *t* = 3.45, *p* = 0.001) had significantly lower density of all stages than observed in the untreated control. 

The density of the feeding stages, only the motiles, in the untreated control treatment ranged between approximately one and 10 mites per leaf in 2016 (Table 1) and one and seven mites per leaf in 2017 (Table 2). Except for one sample date, 22 August 2016 (Table 1), mite densities tended to be at or below approximately one mite per leaf in the preventative acaricide treatments. Only the cyflumetofen treatment had significantly lower mite densities than the untreated control on 25 July (*df* = 21, *t* = 3.07, *p* = 0.027) and 22 August (*df* = 22, *t* = 3.35, *p* = 0.014) of 2016 (Table 1). In 2016 abamectin (*df* = 22, *t* = 4.07, *p* = 0.003), cyflumetofen (*df* = 22, *t* = 4.05, *p* = 0.003), and etoxazole (*df* = 22, *t* = 4.25, *p* = 0.002) each had significantly lower mite densities than the untreated control on 1 August. On 8 August, each of the acaricide treatments, abamectin (*df* = 22, *t* = 4.10, *p* = 0.003), cyflumetofen (*df* = 22, *t* = 4.23, *p* = 0.002), and etoxazole (*df* = 22, *t* = 4.61, *p* < 0.001) had significantly low mite densities than the untreated control.

Densities of the motile stages tended to be numerically greater in the untreated control during 2017 than in the acaricide treatments (Table 2). Only on 21 August were acaricide treatments abamectin (*df* = 20, *t* = 3.80, *p* = 0.006), cyflumetofen (*df* = 20, *t* = 3.20, *p* = 0.021), and etoxazole (*df* = 22, *t* = 4.6, *p* < 0.001) found to be significantly lower than the untreated control (Table 2). No statistical separation of motiles only occurred among the acaricide treatments in either 2016 or 2017 (Table 1 and Table 2). 

Mean mite-days among all treatments ranged between approximately 200 (untreated control) and 41 (cyflumetofen) in 2016; and 132 (untreated control) and 59 (abamectin) in 2017. However, no statistical separation occurred among the treatments during the two-year experiment (*p* > 0.05) Excluding one plot treated with etoxazole in 2016 and a single plot treated with cyflumetofen in 2017, all plots among the acaricide treatments remained below 200 mite-days (Figure 3). In the untreated control, a single plot reached approximately 811 mite-days (2016), and 265 mite-days (2017) (Figure 3).

### 3.2. Spider Mite Predators

We examined approximately 75,330 (2016) and 56,366 (2017) individual leaves during the experiment. Phytoseiid mites or generalist predator insects were not visually detected in either year on leaf samples. In 2017, yellow/blue sticky cards began capturing *S. sexmaculatus* in late spring. Mean ± standard error of the mean (SEM) sixspotted thrips counts ranged between 2.8 ± 2.4 and 12.8 ± 10.8 per card on 16 May, prior to the application of the acaricide treatments. Captures dropped to one or below one thrips per card until 24 July. The highest thrips counts occurred during late August with a high of 40.5 ± 36.5 per card in the abamectin-treated plots (Figure 4). Due to a high level of variation, no statistical difference occurred among the treatments on any of the sample dates (F_3/11_ = 0.72, *p* = 0.5625). 

### 3.3. Susceptible Spider Mites to Abamectin 

An LC_50_ (the lethal concentration which kills 50% of the test population) of abamectin was established for a susceptible population of *T. pacificus* using 455 adult females. Concentrations of, 0.00, 0.23, 0.33, 1.84, and 2.4 ppm were used to evaluate *T. pacificus* and an LC_50_ (95% CL) of 0.39 (0.27–0.52) was determined (Table 3). 

We sampled from each of the experimental sites and six other unique commercial almond orchards located in Fresno, Kern, Kings, and Tulare counties (Table 4) and a single orchard at the University of California, West Side Research and Extension Center in Fresno County between early June and late September of 2017. A subsample of five adult males were taken from each site, slide mounted. Only *T. pacificus* was found in the samples for bioassays.

The model effect for the probit analyses was significant for five of the orchard sample sites, FRSCO1, FRSCO3, FRSCO4, FRSCO5, and TULCO1 (Table 4). The susceptibility of field-collected *T. pacificus* varied considerably with FRSCO1 having a ~3.0-fold and FRSCO4 having an 18.5-fold greater tolerance than the susceptible population (Table 3). The resistant level of the sites can be categorized according to Yan et al. [15] as decreased sensitivity (RR 3.1–5), low resistance (RR 5.1–10), and medium resistant (RR 10.1–40). 

## 4. Discussion

The historically high value of California almonds [16], has encouraged producers to reduce the risk of economic damage i.e., yield loss, which plays a paramount role in their spider mite management program. A preventative acaricide management program aims to protect the almond crop through a critical period of shoot growth and length extension of the pericarp and kernel, which occurs through spring until early June [17]. Additionally, until approximately early August, increases in the dry weight of the nut meat continues [18]. 

This study supports the assertion that the preventative spider mite management strategy does not play a decisive role in suppressing populations to a beneficial degree. We observed over the two-year study that mite populations in treated and untreated plots remained well below economic densities through the critical growth period into late July. In other words, no treatment would be necessary under commercial production practices. Moreover, Welter et al. [3] also found during their two-year study in orchards located in Kern County that the majority of critical growth had occurred prior to spider mites reaching a high level. 

Economic loss can occur if spider mite feeding during those tree growth and nut development stages exceed economic thresholds. Barnes and Andersen [5] determined that heavily infested trees in June through mid-August resulted a yield loss of approximately 16% the following season. Additionally, Welter et al. [3] conducted a study using mite-day measurements and found a similar one-year lag in the reduction of terminal shoot growth and yield. In their study they determined that a significant reduction in leaf size and terminal shoot growth, and yield resulted from 424, 300, and 424 mite-days respectively. Yet, in this study we found that mean cumulative mite-days in untreated control plots remained well below thresholds established by Welter et al. [3].

Although abamectin can have a negative impact on thrips [19,20,21], we did not observe any detectable negative impact on sixspotted thrips during 2017. *Scolothrips sexmaculatus* is an effective predator of tetranychid mites as well as several other mite pests and commonly occurs in North America [17]. Coville et al. [22] reported that at 30 °C, *S. sexmaculatus* requires only 2.7 days to double in population. Moreover, a single female can consume nearly 50 spider mite eggs per day. Additionally, the species is an effective predator at high ambient temperatures making it well suited for California’s almond production regions and has been recorded as an effective spider mite predator on almond during the latter part of the growing season [2]. The precipitous decline in the mites and eggs in 2017 coincided with the increasing presence of *S. sexmaculatus* captures during late summer and likely played a role in the precipitous decrease observed in this experiment. 

The California Department of Pesticide Regulation (CDPR) reported that between 2012 and 2017 approximately half the almond hectares received at least one acaricide application at the preventative management timing of May. Moreover, abamectin accounted for approximately 90% of applications [6]. Although, abamectin has the most efficacy prior to leaves hardening off [6], the acaricide also is heavily relied upon as later-season treatments. The summation of hectares treated with at least one abamectin application during the months of June, July, August, and September totaled 158, 196, and 239 hundred thousand for 2015, 2016 and 2017, respectively [6]. The commercial orchards used in this study typically received a single label rate application of abamectin at the April or May timing, and a second application of ether etoxazole (Zeal), fenpyroximate (Fujimite 5EC), hexythiazox (Onager) or pymtrozine (Envidor) prior to hull split. 

Mites in the genus *Tetranychus* have a well-documented history of developing acaricide resistance [8,23]. In *T. urticae*, for example, Ferreira et al. [24] estimated resistance ratios of 8.0 to 295,270 in populations collected in Brazil from cotton and ornamental flower plantations. On ornamental nursery plants produced in California, Campos et al. [25] reported resistance ratios for *T. urticae* from 5 to 87. Also, Zalom [26] evaluated populations of *T. urticae* on strawberry from Orange, Ventura, Santa Barbara, San Luis Obispo, and Monterey counties and found populations between 125- and 1738-fold more resistant than a susceptible population.

We did not obtain spray records from the anonymous orchard sites; however, given the resistant ratios calculated, four of the populations have developed low to medium resistance. The LC_50_ ppm values determined in this study did not exceed the Agri-Mek SC recommended field rate (4.4–8.7 g AI/Ac or 11.5–23 ppm). Our results support reports by almond producers and pest control advisers (PCAs) working in Central Valley nut crops that abamectin efficacy has diminished over the past decade.

Abamectin resistance is unstable in the absence of selection [27,28]. For instance, Brown et al. [29], found that on cotton large shifts in *T. urticae* susceptibility occurred from year to year. They concluded that enough time elapses between seasons that susceptibility can reestablish. This characteristic of abamectin is beneficial in spider mite management strategies [30]. Given the low level of resistance we found during this study, abamectin should remain an effective tool as long as growers and PCAs take resistance management seriously and practice sound IPM of spider mites on almond.

## 5. Conclusions

Our results indicated that the preventative mite treatments provided mite suppression over a three-week period (late July to early Aug) of year one, although not during year two of the experiment. We observed, however, that spider mite populations in all treated and untreated plots remained well below economic levels through the growth period critical for yield. Moreover, at that period in the season, growers reduce irrigation and try to avoid applying insecticides or acaricides due to the temporal proximity of the harvest, in other words the preventative spray did not provide an economic benefit. Additionally, populations were present only for approximately five weeks and, therefore, mite feeding injury, or mite-days remained well below damage thresholds.

In this study, we showed that a medium level of resistance to abamectin has developed in some Pacific mite populations in the south and mid-San Joaquin Valley. Abamectin is an important tool for managing spider mites. To maintain the acaricide’s effectiveness, the almond industry should work to eliminate the practice of making preventative applications and rely more on monitoring during May for making early-season treatment decisions. The industry must work toward a greater reliance on sixspotted thrips. The predator likely plays a role in managing early-season mite populations and, as the results of this study suggests, plays a critical role in managing spider mites at late-season.

## Figures and Tables

**Figure 1 insects-11-00772-f001:**
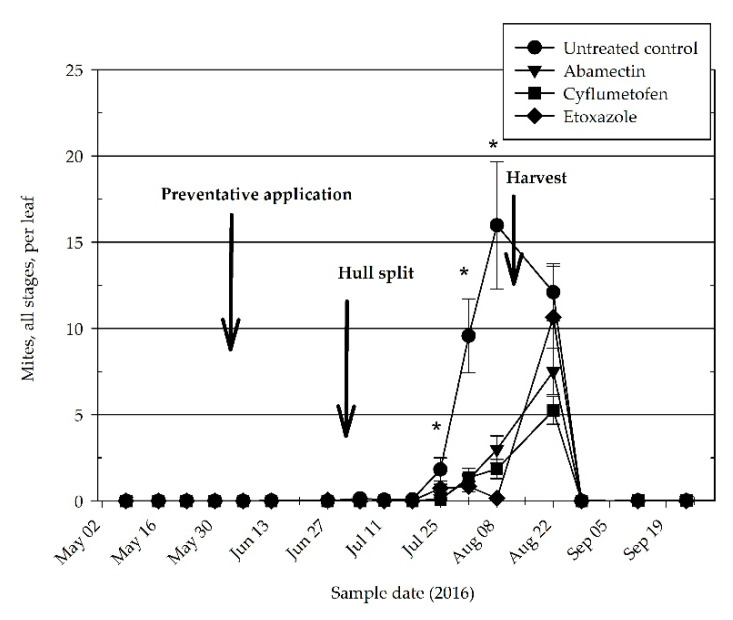
Density per leaf (±standard error of the mean (SEM)) of spider mites (all stages) on almond in 2016. Data pooled from the two orchard sites located near Shafter and Wasco, Kern County. Asterisk indicates mean spider mite density is significantly greater in untreated control than at least one preventative acaricide treatment.

**Figure 2 insects-11-00772-f002:**
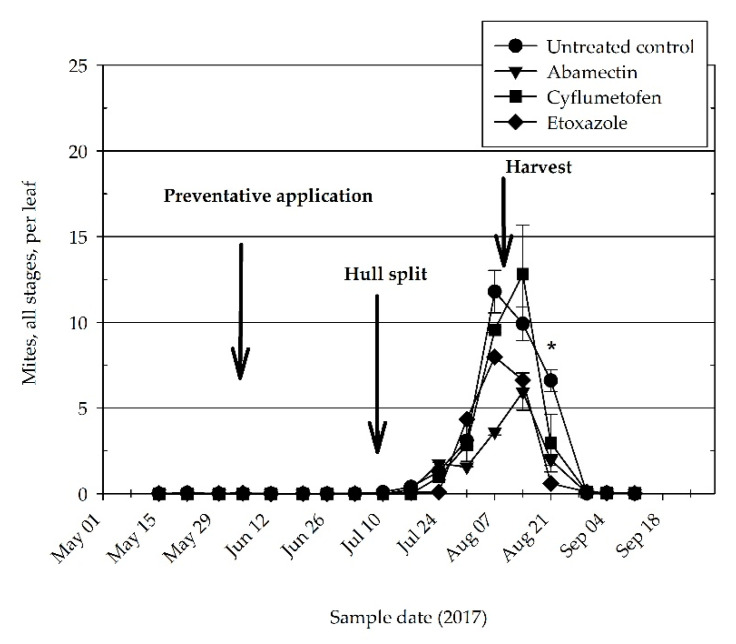
Density per leaf (±SEM) of spider mites (all stages) on almond in 2017. Data pooled from the two orchard experimental sites located near Shafter and Wasco, Kern County. Asterisk indicates mean spider mite density is significantly greater in untreated control than at least one preventative acaricide treatment.

**Figure 3 insects-11-00772-f003:**
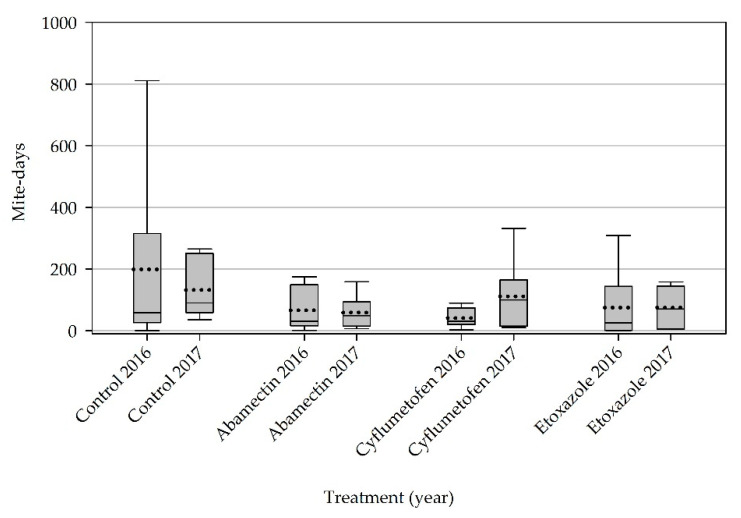
Cumulative mite-days. Data pooled from two experimental orchard sites located near Shafter and Wasco, Kern County. The box of the Box-whisker plot indicates the 25% and 75% percentile range of mean mite-days. The vertical line indicates the minimum and maximum values. The horizontal dash line and solid line in each box indicate the mean and median mite-days respectively.

**Figure 4 insects-11-00772-f004:**
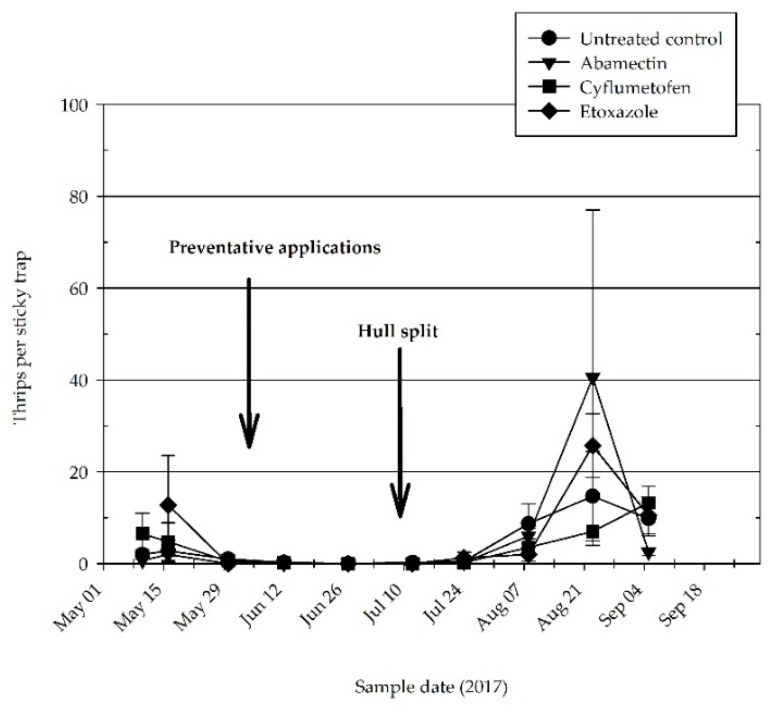
Mean (±SEM) density of sixspotted thrips per leaf on almond 2017. Data pooled from two experimental orchard sites located near Shafter and Wasco, Kern County.

**Table 1 insects-11-00772-t001:** Number of mites per leaf on almond treated with various acaricides in 2016.

	Mean Spider Mites ± SEM Per Leaf			
Date	Untreated Control	Abamectin	Cyflumetofen	Etoxazole
7/25	0.79 ± 0.27 a	0.06 ± 0.04 ab	0.02 ± 0.01 b	0.39 ± 0.13 ab
8/1	5.64 ± 1.40 a	0.70 ± 0.17 b	0.73 ± 0.35 b	0.48 ± 0.16 b
8/8	9.68 ± 2.44 a	1.13 ± 0.29 b	0.85 ± 0.28 b	0.07 ± 0.03 b
8/15 ^1^				
8/22	6.16 ± 0.86 a	3.80 ± 0.67 ab	2.15 ± 0.35 b	4.95 ± 1.28 ab

Means in rows with the same letter are not significantly different. ^1^ Samples not collected due to harvest.

**Table 2 insects-11-00772-t002:** Number of mites per leaf on almond treated with various acaricides in 2017.

	Mean Spider Mites ± SEM Per Leaf			
Date	Untreated Control	Abamectin	Cyflumetofen	Etoxazole
7/24	0.67 ± 0.17 a	0.75 ± 0.37 a	0.38 ± 0.13 a	0.03 ± 0.02 a
7/31	1.18 ± 0.31 a	0.58 ± 0.18 a	1.12 ± 0.41 a	1.40 ± 0.36 a
8/7	7.27 ± 1.72 a	2.38 ± 0.65 a	5.87 ± 2.10 a	5.00 ± 0.98 a
8/14	5.31 ± 0.93 a	3.75 ± 0.64 a	6.94 ± 1.11 a	3.86 ± 0.87 a
8/21	4.27 ± 0.69 a	1.05 ± 0.48 b	1.55 ± 0.52 b	0.32 ± 0.10 b

Means in rows with the same letter are not significantly different.

**Table 3 insects-11-00772-t003:** Concentration response of susceptible and field-collected populations of *Tetranychus pacificus* to abamectin after 24 h of exposure. Bioassays conducted in 2017.

Population	N ^1^	Slope log_10_(ppm) (SE)	Χ^2^ (*p* Value)	LC_50_ (95% CL), ppm	Resistance Ratio ^2^
SUS1	455	1.33 (0.17)	59.80 (<0.0001)	0.39 (0.27–0.52)	-
FRSCO1	945	1.50 (0.16)	84.10 (<0.0001)	1.16 (0.98–1.14)	3 (0.95–9.43)
FRSCO2	189		2.10 (0.1498)		
FRSCO3	467	0.72 (0.14)	27.90 (<0.0001)	6.24 (3.63–12.77)	16 (3.88–65.91)
FRSCO4	617	1.07 (0.18)	33.80 (<0.0001)	7.23 (4.7–14.26)	18.5 (3.69–92.84)
FRSCO5	791	0.81 (0.12)	42.20 (<0.0001)	2.28 (1.51–3.55)	5.8 (1.49–22.52)
KERCO1	381		0.59 (0.4415)		
KERCO2	477		1.68 (0.1948)		
KERCO3	250		0.24 (0.2444)		
TULCO1	376	0.53 (0.21)	6.71 (<0.0096)	5.11 (1.83–2375)	13.1 (2.37–72.54)

^1^ Number of adult females used in the bioassays, including controls. ^2^ Resistance ratio calculated as (LC_50_ of field population/ LC_50_ susceptible population). 95% CL calculated according to Robertson et al. [14].

**Table 4 insects-11-00772-t004:** Code and location of spider mite sample sites in 2017.

Code	Location
SUS1	University of California, Kearney Ag Station
FRSCO1	Navelencia, Fresno Co
FRSCO2	Fresno, Fresno Co
FRSCO3	Raisin City, Fresno Co
FRSCO4	University of California, Westside Field Station
FRSCO5	West Manning Ave
KERCO1	McFarland, Kern Co
KERCO2	Schofield, Kern Co (experimental site 1)
KEROC3	Whistler, Kern Co (experimental site 2)
TULCO1	Corcoran, Tulare Co

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
