# Peer review of "Evaluation of a ‘Preventative’ Strategy to Manage Spider Mites on Almond"

_insects, 2020, doi:10.3390/insects11110772_

Round 1

Reviewer 1 Report

Overall, the paper is much more concise has addressed many of the issues brought up in the first review. However, references need to be checked. Brown et al [23]  was not included in the references(Line 483). Reference number [9] is cited as Mound in the text and A., M.L. in the references (line 99). Robertson et al [14] is incorrect in table 3 (line 395). Citation [19] is not included in the paper (line 558). Citation [20] is not included in the paper (line 560). Citation [24] is not included in the paper (line 571). 

Author Response

Overall, the paper is much more concise has addressed many of the issues brought up in the first review. However, references need to be checked. Brown et al [23] was not included in the references(Line 483). Reference number [9] is cited as Mound in the text and A., M.L. in the references (line 99). Robertson et al [14] is incorrect in table 3 (line 395). Citation [19] is not included in the paper (line 558). Citation [20] is not included in the paper (line 560). Citation [24] is not included in the paper (line 571).
Comments and Responses
Comment: Brown et al. [23] not included
Response: Brown reference corrected
Comment: Reference number [9] is cited as Mound in the text and A., M.L. in the references (line 99).
Comment: Response: Mound corrected in citations
Comment: Robertson et al [14] is incorrect in table 3 (line 395). Citation [19] is not included in the paper (line 558). Citation [20] is not included in the paper (line 560). Citation [24] is not included in the paper (line 571).
Response: All references were actually in the text. I looked over text and doublechecked references.
Thank you reviewer 1 for you constructive comments.

Reviewer 2 Report

This paper reports field trials on the control of spider mites in almonds in California. The objective is to check of the so-called preventive strategy is effective. This is important research as it contributes to the reduction of the use of pesticides on an important crop grown on a large area.

The field trials were carried out well and sampling, data collection and statistics are done according to current standards. I would however ask the authors to report d.f., F, and p values in the results (see e.g. Insects 2020, 11, 670; doi:10.3390/insects11100670 as an example.

What is the reason for reporting motile stages + eggs and motile stages only separately (and motiles + eggs in figures and motiles only in tables)?. In a trial like this (applied research in commercial orchards) I would only report the total population. Or are there differences in the effectiveness of the acaricides used against eggs and motile stages or any other reason to report the data like this? Then this should be indicated and discussed.

I prefer the use of the term acaricde instead of miticide.

Abstract

Mention that only T. pacificus was found in the abstract when starting with reporting results (L 14/15)

Generally, compared to the rest of the ms, the abstract is not written well.

L 17 … mite densities -… tended to have more mites should be tended to be higher

L 19 (and the entire manuscript) use significantly different instead of statistically different

L 22 typing error: Pergande is correct

L 23 replace chemistry by product or active ingredient

L 25 Typing error: Tetranychus is correct

Introduction

L 53 I think it is 32% in absence of predators and 53% in presence. As it is written now it looks as if it is the other way round

L 54 add measure after preventative

L 57 95-68%, what do you mean?

L 61 what is an economic population?

Materials & Methods

L 79 KpH, please use the SI symbol: km/h

L 80 late spring: please indicate (range of) dates

L 137 150 x 15 diameter x height?

L 143-149 Eggs and motiles were sampled and are presented in the results, including statistical differences. Here only a procedure for motile stages is described

L 148 typing error: should be least square-means

Results

L 170 significantly lower than what?

As said above, I do not quite understand why data for total population and motiles only are presented separately (and the eggs not)

L 187-192 Figure 3 does not show means if I understand correctly

L 198 are these means ± SEM? one decimal digit is enough (as in line 200)

L 216 use either 3 and 19 or 3.0 and 18.5

Figure legends, Fig. 1 & 2: be more clear, either use all stages or spider mite motile stages plus eggs. What do the asterisks indicate in the figures?

Figure 3: What does this figure show, what are the boxes and whiskers? For a good figure legend of such a graph see e.g. Dively et al, Insects 2020, 11, 614

Figure 4. Legend: according to the result section no thrips were found on leaves. Should be sticky traps, I think. And what are the data: means, SEM?

Tables 1 and 2: indicate clearly what is shown, now the legend indicates “mites per leaf” and the table header “spider mites stages per leaf”, what is UTC? here SE, earlier SEM

Table 3: Be consistent with number of decimal digits, P value for FRSCO5 is missing, reference 14 is not Robertson et al.

Discussion

L 444 mite days was already explained earlier, not necessary here again

L 460-469 Are these figures and values correct? According to the introduction there are 0.45 mio ha of almonds in CA, half of them received a preventive treatment and 90% abamectin. This does not fit with 158, 196 and 239 ha. How can a publication of 2016 (7) contain data from 2017?

L 474 the reference for Zalom is missing

Conclusions

L 491-497 This paragraph is not clear to me. You indicate that there was a reduction in the first year but not in the second. This is clear and in accordance with the data. Furthermore, the populations remained below the economic threshold also in untreated. Therefore, even if there was an effect, the treatments were obsolete, correct? This should be stated more clearly.

Thereafter, it gets cofusing. Growers do not treat after late July? This is when problems could occur. Then they cannot do anything? Why to monitor in August (see introduction) if they cannot treat anyway? What is the significance of the 5 weeks? Did the populations remain below the threshold for the entire season in both years? Please reformulate this ppragraph to make it more clear. In case the population remained always below the threshold, not only preventative treatments are obsolete but all acaricide treatments are obsolete. This should then be stated clearly.

Author Response

Comments and Responses

This paper reports field trials on the control of spider mites in almonds in California. The objective is to check of the so-called preventive strategy is effective. This is important research as it contributes to the reduction of the use of pesticides on an important crop grown on a large area.
The field trials were carried out well and sampling, data collection and statistics are done according to current standards.
Comment: I would however ask the authors to report d.f., F, and p values in the results (see e.g. Insects 2020, 11, 670; doi:10.3390/insects11100670 as an example.
Response: d.f, F, and P values provided in the text for all the needed comparisons.

Comment: What is the reason for reporting motile stages + eggs and motile stages only separately (and motiles + eggs in figures and motiles only in tables)?. In a trial like this (applied research in commercial orchards) I would only report the total population. Or are there differences in the effectiveness of the acaricides used against eggs and motile stages or any other reason to report the data like this? Then this should be indicated and discussed.
Response: The text at line 68 states that the University of California developed a system using both eggs and motile stages to assess populations and make a treatment decision. We feel that the statement suffices in explaining why both all stages (Figure 1 and 2) and only the feeding stage i.e. motiles were reported separately. The motile stages are of course the feeding stage and are important for determining actually feeding injury, mite-days.

Comment: I prefer the use of the term acaricde instead of miticide.
Response: Acaricide replaced miticide throughout manuscript.

Abstract
Comment: Mention that only T. pacificus was found in the abstract when starting with reporting results (L 14/15)
Response: Text edited to read that T. pacificus was the only species identified during the study.
Comment: Generally, compared to the rest of the ms, the abstract is not written well.
Response: I have rewritten the abstract to clarify
Comment: L 17 … mite densities -… tended to have more mites should be tended to be higher
Response: Higher added to text.
Comment: L 19 (and the entire manuscript) use significantly different instead of statistically different
Response: significantly replaced statistically throughout text
Comment: L 22 typing error: Pergande is correct
Response: I did not see that this was incorrect?
Comment: L 23 replace chemistry by product or active ingredient
Response: Chemistry replaced with active ingredient.
Comment: L 25 Typing error: Tetranychus is correct
Response: Tetranychus spelling error corrected.

Introduction
Comment: L 53 I think it is 32% in absence of predators and 53% in presence. As it is written now it looks as if it is the other way round
Response: Comment: Yes incorrect as indicated by Reviewer 2. I have corrected the error.
Comment: L 54 add measure after preventative.
Response: Measure added.
Comment: L 57 95-68%, what do you mean?
Response: Edited text to clarify meaning of
Comment: L 61 what is an economic population?
Response: Economic damage added to text to clarify meaning.

Materials & Methods
Comment: L 79 KpH, please use the SI symbol: km/h
Response: Corrected
Comment: L 80 late spring: please indicate (range of) dates.
Response: Range of application date added as late May.
Comment: L 137 150 x 15 diameter x height?
Response: diameter and height added to clarify dimension.
Comment: L 143-149 Eggs and motiles were sampled and are presented in the results, including statistical differences. Here only a procedure for motile stages is described.
Response: The text has been edited to clarify that all mite stages and motile stage only were analyzed.
Comment: L 148 typing error: should be least square-means
Response: Least squares means is correct. Least squares means is a SAS output. It is defined as the sum of the means from a linear model.

Results
Comment: L 170 significantly lower than what?
Response: Significantly lower than control added to text.
Comment: As said above, I do not quite understand why data for total population and motiles only are presented separately (and the eggs not)
Response: The text at line 68 states that the University of California developed a system using both eggs and motile stages to assess populations and make a treatment decision. We feel that the statement suffices in explaining why both all stages (Figure 1 and 2) and only the feeding stage i.e. motiles were reported separately. The motile stages are of course the feeding stage and are important for determining actually feeding injury, mite-days
Comment: L 187-192 Figure 3 does not show means if I understand correctly
Response: Clarification added to the figure legend to indicate that the broken line in the box indicates means.
Comment: L 198 are these means ± SEM? one decimal digit is enough (as in line 200)
Response: SEM added decimal digit edited as per comment.
Comment: L 216 use either 3 and 19 or 3.0 and 18.5
Response: 3.0 added to be consistent with 18.5.
Comment: Figure legends, Fig. 1 & 2: be more clear, either use all stages or spider mite motile stages plus eggs. What do the asterisks indicate in the figures?
Response: All stages added to legend of figures one and 2. Asterisk defined in the legend.
Comment: Figure 3: What does this figure show, what are the boxes and whiskers? For a good figure legend of such a graph see e.g. Dively et al, Insects 2020, 11, 614
Response: Added text to explain the box-whisker plot in the legend. Reference given by Reviewer 2 used as a guide.
Comment: Figure 4. Legend: according to the result section no thrips were found on leaves. Should be sticky traps, I think. And what are the data: means, SEM?
Response: On leaf samples is correct. Thrips were found only on sticky traps. SEM added to Figure 4 legend.
Comment: Tables 1 and 2: indicate clearly what is shown, now the legend indicates “mites per leaf” and the table header “spider mites stages per leaf”, what is UTC? here SE, earlier SEM
Response: Text in the tables edited to read means mites per leaf ± SEM. Untreated control replaced UTC in tables 1 and 2.
Comment: Table 3: Be consistent with number of decimal digits, P value for FRSCO5 is missing, reference 14 is not Robertson et al.
Response: inconsistency corrected and P value added for FRSCO5 in table. Reference for Roberson et al . corrected to read [14]。

Discussion
Comment: L 444 mite days was already explained earlier, not necessary here again
Response: Explanation for mite-days removed.
Comment: L 460-469 Are these figures and values correct? According to the introduction there are 0.45 mio ha of almonds in CA, half of them received a preventive treatment and 90% abamectin. This does not fit with 158, 196 and 239 ha. How can a publication of 2016 (7) contain data from 2017?
Response: No the numbers were not correct. The numbers represent hundred thousands. Text corrected to clarify that the values are hundred thousands.
Comment: L 474 the reference for Zalom is missing
Response: Zalom added.

Conclusions
Comment: L 491-497 This paragraph is not clear to me. You indicate that there was a reduction in the first year but not in the second. This is clear and in accordance with the data. Furthermore, the populations remained
below the economic threshold also in untreated. Therefore, even if there was an effect, the treatments were obsolete, correct? This should be stated more clearly.
Thereafter, it gets cofusing. Growers do not treat after late July? This is when problems could occur. Then they cannot do anything? Why to monitor in August (see introduction) if they cannot treat anyway? What is the significance of the 5 weeks? Did the populations remain below the threshold for the entire season in both years? Please reformulate this ppragraph to make it more clear. In case the population remained always below the threshold, not only preventative treatments are obsolete but all acaricide treatments are obsolete. This should then be stated clearly.
Response: I have addressed these concerns and have edited the conclusion to better explain the conclusion of the trial. Albeit, there is no definitive results. What we have shown is that growers need not spray preventative treatments because the mites do not show up until after the period critical for tree growth and yield. But it does not mean that monitoring should stop and that a treatment could not be applied just because it is close to harvest. There are some acaricides that have a low preharvest interval that could be used, but growers tend to avoid conflicts with harvest timing.

Reviewer 3 Report

Review of Tollerup and Higbee

This is a generally well written study evaluating the common practice of the almond industry to apply a preventative measure to control mites, which is inconsistent with CA IPM program’s recommendation to apply a miticide only when 32% or 53% of leaves are infested.

Results are inconclusive in that 2016 and 2017 results differed, although it can be argued that in both years mite numbers eventually increased to control levels. This would justify the authors’ conclusion on L501 that the preventative strategy be eliminated.

The paper addresses (1) preventative strategy; (2) resistance to abamectin; (3) natural enemies (thrips). Need to tie the three subjects together in the Intro a little better.

In this regard, I would place L50-58 (strategy) before L45-49 (natural enemies) and then devote a paragraph including L61-63 on resistance before L59. Also natural enemies and resistance to the strategy.

There are editorial errors that authors should correct, but I will let authors make those minor edits.

-------------------------------

Title: Should include some mention of testing for mite resistance to abamectin, since this was a major part of the study; maybe even the mention of sixspotted thrips somehow.

Abstract

Needs a concluding sentence tying the findings together rather than all results.

Essentially, is it concluded that the strategy is effective for managing the mite and preventing damage to almonds. Tough here since in 2016 (Table 1) there is evidence for it being effective but then in 2017 (Table 2) there is no evidence for it being effective.

The timing of when mites appear is key, as it occurs after the critical period of almond (L430), so this should be somehow be in the abstract.

Lines 61-65. Make a better connection between preventative miticide applications and resistance – how it can relate to the preventative strategy. L500-502 may be a lead.

L79: state dates when applied. Fig. 2 shows about 30 May 2016 and about 11 June 2017 and these dates should be stated here.

L90: placed

L130: state how many replicates

Discussion in General

Make it clear that the preventative miticide applications could suppress mite populations (2016) but are not necessary as mites appear after the critical period and that mites will appear later in the season.

L491-497: There is no real conclusion here, just results. Evidence for efficacy of the strategy in one year, but not in the next year. Happens a lot in field studies of this sort. Maybe a third year if study to tilt conclusions one way?

Author Response

Comment: Title: Should include some mention of testing for mite resistance to abamectin, since this was a major part of the study; maybe even the mention of sixspotted thrips somehow.
Response: Thank you for the suggestion, however, we have decided to not add abamectin to the title.

Abstract
Comment: Needs a concluding sentence tying the findings together rather than all results.
Response: We have rewriting the abstract to clarify and tie the results together.

Comment: Essentially, is it concluded that the strategy is effective for managing the mite and preventing damage to almonds. Tough here since in 2016 (Table 1) there is evidence for it being effective but then in 2017 (Table 2) there is no evidence for it being effective.
Response: No, I have attempted to stress that preventative sprays had some effect in 2016 but the critical period for tree growth and nut yield have passed without mites being present. This is typical as per other studies Walter et al 1984.
Comment: The timing of when mites appear is key, as it occurs after the critical period of almond (L430), so this should be somehow be in the abstract.
Response: With all due respect, I disagree. The greater details of the study should be left for the body of the manuscript.
Comment: Lines 61-65. Make a better connection between preventative miticide applications and resistance – how it can relate to the preventative strategy. L500-502 may be a lead.
Response: I have edited the introduction to make a better connection between abamectin as a preventative spray.
Comment: L79: state dates when applied. Fig. 2 shows about 30 May 2016 and about 11 June 2017 and these dates should be stated here.
Response: Date range indicating late May added to text.
Comment: L90: placed
Response: Corrected
Comment: L130: state how many replicates
Response: Number of replicates stated.

Discussion in General
Comment: Make it clear that the preventative miticide applications could suppress mite populations (2016) but are not necessary as mites appear after the critical period and that mites will appear later in the season.
Response: In the general discussion, I have rewritten parts of the discussion to stress that mite suppression may not needed until very late in the season.
Comment: L491-497: There is no real conclusion here, just results. Evidence for efficacy of the strategy in one year, but not in the next year. Happens a lot in field studies of this sort. Maybe a third year if study to tilt conclusions one way?
Response: with all due respect, we believe that this two-year study provides sufficient data to discount the need for preventative acaricide applications.

This manuscript is a resubmission of an earlier submission. The following is a list of the peer review reports and author responses from that submission.

Round 1

Reviewer 1 Report

I was not able to finish my review of the paper once I made it to the data portion. The statistical methods were not clearly outlined and hard to follow. No means separation analysis was given and why the pdiff option was used instead of Tdiff (especially since the authors stated they were comparing t-test outcomes). The experimental methods need to be clearly defined and the paper needs to be reanalyzed before resubmission. My recommendation is to reconsider after major revision. My review includes corrections up to the first table line 227. 

Reviewer 2 Report

SUMMARY

The authors conducted field surveys in order to determine whether the prophylactic use of three pesticides (including abamectin) reduces mite populations in almond orchards. They further investigated if mites developed resistance. They concluded that treatments did not provide management benefits. Moreover, field mites are more tolerant to abamectin compared to laboratory maintained, susceptible ones.

BROAD COMMENTS

It is not clear from the ‘Material and Method’ and ‘Results’ sections what the diversity of mite species was in the field. Looking at figures and tables suggests that total numbers of individuals were reported, no matter the species they belong to. As a consequence, it is not possible to relate the field experiments to the laboratory bioassays. If a single mite species was harvested in the field during that study, this has to be explicit. If there was several of them, then results per species should be provided.

Investigating the population of a single potential predator, as the authors did, does not allow concluding about its contribution to pest management. I think they have to lower the tone in that respect.

Species names should be italicised.

SPECIFIC COMMENTS

L11-13: The sentence could be more specific. Effectiveness in what? L14: The species name for ‘pacific mite’ should be mentioned. L16: How is a ‘significant level’ defined? L18: The species name for ‘sixspotted thrips’ should be mentioned. L18: That thrips impacted mite populations is suggested by the data, not demonstrated. The sentence has to be nuanced in that respect. L46-48: References are required. L64-65: Objectives should be more specific. In what respect were preventive applications evaluated? L67: What do the authors mean by ‘and/or’? L71-77: A figure would be welcome. L71: What was the treatment history of the plots? What was the distance between them? Could one plot be contaminated during the treatment of another? L73-74: Was it applied in both 2016 and 2017? Was it the same treatment in the same plots over the two years? L87: Was the distinction between species made? How was performed identification? L102: It is ‘2.4’. L104: Was there a target mite species? L112: It is ‘2.5’. L114: What did the ‘untreated control’ consist in? L117: What were the test concentrations? L122-124: What volume had to be sprayed per disc to cover its surface? L129: What was the total duration of a bioassay? L130: It is ‘2.6’. L201: What ‘LD50’ stands for should be explained. Also, ‘LD50’ or ‘LC50’? L201: I do not think T. urticae was mentioned before, L97-101. L210: Table 4 does not provide any insight about significance. L212-214: How impartial is that? How the authors define resistance? I would delete. L227: I am actually surprised by significance indices in Table 1. How comes, for instance, that UTC in SMtot (8/22) is significantly different from etoxazole but not from abamectin? I agree that variance is a bit higher in there, but still. Do the authors have an explanation? What was the sampling size for the different items? L268: Table 4 should come after Table 3. L272: Title: There is not only T. pacificus in that table. L272: How much time after exposure? Plus, it is not possible to judge if significant or not. No more statistical estimates? What about intercept, degrees of freedom and p-values? L289: see comment L16. L331: Passive form is much more common in scientific reports. I would rephrase. L338: Second… L338-339: See comment L18. This is suggested, not demonstrated. L346: Same comment. L351: See comment L331.

Reviewer 3 Report

The purpose of this study is not clear. I was uncomfortable when I read that the researchers were evaluating a prophylactic/calendar-based spray because I was concerned that growers might take this as evidence that calendar-based sprays are okay. I expected to read in the paper that the authors intended to show growers why calendar-based sprays are not a good idea, but this was not emphasized. The research design doesn't seem to make much sense. It would have made sense to compare one or more calendar-based sprays to one or more IPM (scouting-based) programs. But the authors just compared treatments where applications were made without regard to mite thresholds. Presumably thresholds have been established for mites in California almonds, but this information is not shared or incorporated into the research design. The manuscript clearly represents a lot of work, but it was not well thought out.